# Efficiency of Antimicrobial Photodynamic Therapy with Photodithazine^®^ on MSSA and MRSA Strains

**DOI:** 10.3390/antibiotics10070869

**Published:** 2021-07-17

**Authors:** Beatriz Müller Nunes Souza, Juliana Guerra Pinto, André Henrique Correia Pereira, Alejandro Guillermo Miñán, Juliana Ferreira-Strixino

**Affiliations:** 1Laboratory of Photobiology Applied to Health, Research and Development Institute, University of Vale do Paraíba, Urbanova 2911, Brazil; beatriz97.muller@yahoo.com.br (B.M.N.S.); juguerra@univap.br (J.G.P.); andre_gcp@hotmail.com (A.H.C.P.); 2Instituto de Investigaciones Fisicoquímicas Teóricas y Aplicadas, Facultad de Ciencias Exactas, Universidad Nacional de La Plata, La Plata 1900, Argentina; agminan@inifta.unlp.edu.ar

**Keywords:** antimicrobial photodynamic therapy, MRSA, MSSA, photodithazine^®^

## Abstract

*Staphylococccus aureus* is a ubiquitous and opportunistic bacteria associated with high mortality rates. Antimicrobial photodynamic therapy (aPDT) is based on the application of a light source and a photosensitizer that can interact with molecular oxygen, forming Reactive Oxygen Species (ROS) that result in bacterial inactivation. This study aimed to analyze, in vitro, the action of aPDT with Photodithazine^®^ (PDZ) in methicillin-sensitive *Staphylococcus aureus* (MSSA) and methicillin-resistant *Staphylococcus aureus* (MRSA) strains. The strains were incubated with PDZ at 25, 50, 75, and 100 mg/L for 15 min and irradiated with fluences of 25, 50, and 100 J/cm^2^. The internalization of PDZ was evaluated by confocal microscopy, the bacterial growth by counting the number of colony-forming units, as well as the bacterial metabolic activity post-aPDT and the production of ROS. In both strains, the photosensitizer was internalized; the production of ROS increased when the aPDT was applied; there was a bacterial reduction compared to the control at all the evaluated fluences and concentrations; and, in most parameters, it was obtained complete inactivation with significant difference (*p* < 0.05). The implementation of aPDT with PDZ in clinical strains of *S. aureus* has resulted in its complete inactivation, including the MRSA strains.

## 1. Introduction

*Staphylococcus aureus* is a Gram-positive bacterium responsible for a wide variety of pathologies, from skin diseases, such as acne, furuncle, pustules, and impetigo to even more serious ones, such as pneumonia, osteomyelitis, endocarditis, meningitis, and sepsis [1,2]. This pathogen is often found as a contaminant in hospital environments and materials, besides being a frequent hindrance in injuries such as burns and skin abscesses in hospitalized patients. In 2016, a study by ANVISA (Agência Nacional de Vigilância Sanitária) showed that *S. aureus* is among the five microorganisms with the highest incidence in hospitalized patients in the adult (14.1%) and pediatric (11.5%) intensive care units [3,4].

In recent years, antimicrobial resistance has become an alarming issue worldwide. Forecasts indicate deaths of 10 million people a year, accrued expenses of 1 trillion dollars, and a reduction of the Gross Domestic Product of up to 3%, especially in emerging countries. In 2017, the World Health Organization (WHO) classified methicillin-resistant *Staphylococcus aureus* (MRSA) and vancomycin-resistant *S. aureus* (VRSA) as high-priority pathogens, requiring global efforts to develop new treatments and drugs to mitigate the future consequences of infections by these superbugs [5,6,7]. In 2019, WHO continually warns of the increase in resistant species and the difficulty of developing efficient antibiotics for these pathogens. Currently, only seven antibiotics are in the pre-clinical phase for the treatment of MRSA infections [8].

Due to the increasing bacterial resistance, the number of antibiotics described in the literature is limited, and since there is no development forecast of new drugs, alternative treatments are required to control bacterial infections. The aPDT employs a photosensitizer (PS) that is excited by the light emitted at a particular wavelength by a light source, after being internalized into the cell. In this process, it interacts with molecular oxygen producing highly toxic Reactive Oxygen Species (ROS) that result in bacterial inactivation [9,10,11]. In this sense, aPDT has been suggested as an efficient alternative approach to be applied to the management of clinical contact surfaces, disinfection of surgical instruments, biofouling, and even antimicrobial treatment of wastewater [12].

Chlorins are second-generation PS formed from the reduction of a porphyrin pyrrole ring. This PS presents antimicrobial activity at the 630–660 nm range of the visible light spectrum. It presents a high ROS yield, low toxicity in the dark, and is employed in the treatment of superficial and deep tissue injuries due to the high penetration capacity of the light [13,14,15]. Photodithazine^®^ (PDZ) is an e6 chlorine of Russian origin, obtained from the cyanobacterium *Spirulina platensis* which has shown promising results in aPDT [16,17,18,19]. The objective of the present study was to evaluate the internalization of PDZ and the action of aPDT with PDZ on the viability, metabolism, and production of ROS in the MRSA and MSSA strains.

## 2. Results

### 2.1. Internalization of PDZ in the MSSA and MRSA Strains with Confocal Microscopy

The results obtained by confocal microscopy after 15 min of incubation with PDZ at 100, 75, 50, and 25 mg/L allowed the observation of the bacterial morphology characteristic of *Staphylococcus* spp. in grape-like clusters arrangements. At all the concentrations evaluated, PDZ interacted with the bacterial cells allowing us to see that the PDZ fluorescence had filled the whole bacterial morphology, suggesting internalization. In addition, there were no visible differences in internalization between MRSA and MSSA strains, as well as between the different PDZ concentrations, suggesting that the concentration of 25 mg/L with an incubation time of 15 min was sufficient for internalization, as shown in Figure 1 and Figure 2.

### 2.2. Evaluation of Bacterial Growth by Counting the Number of Colony Forming Units (CFU).

In the MSSA strains, a reduction in growth (*p* < 0.05) was observed among the aPDT groups in relation to the control, demonstrating that the therapy was effective at all PS concentrations and all light intensities (Figure 3). In addition, there were no statistical differences between the control group and the non-irradiated PDZ groups, proving that PDZ did not exhibit toxicity in the dark. The groups that were only irradiated with different fluences did not show any statistical differences in growth compared to the control group (*p* > 0.05).

The use of PDZ therapy reduced bacterial growth, resulting in bacterial eradication in most of the tested parameters, even with the sensitivity of the strains to the antibiotic methicillin. The exception was the application of the therapy at 50 and 25 mg/L, combined with a fluence of 25 J/cm^2^ and concentration of 25 mg/L, and with a fluence of 50 J/cm^2^, which did not cause complete bacterial inactivation, concluding that the low fluence and PDZ concentrations are not suitable for complete inactivation of the MSSA strains. The effect of the 75 mg/L concentration on all applied irradiations stands out, suggesting it is a great alternative to achieve bacterial eradication.

The result of aPDT for the MRSA strains (Figure 4) was similar to that of the MSSA strains, resulting in total bacterial inactivation in most of the groups tested, except at 25 mg/L of PDZ, combined with a fluence of 25 J/cm^2^, which resulted in a difference of approximately 5 logs compared to the non-irradiated control group. In this sense, it is important to note that even using the lowest PS concentration (25 mg/L) together with the lowest fluence (25 J/cm^2^), a bactericidal effect (decreased viability ≥3 log) was observed in both the susceptible bacteria as well as in multidrug-resistant strains. The fact that most of the tested concentrations and fluences resulted in total bacterial inactivation demonstrates, in this case, the effectiveness of aPDT, regardless of the antibiotic resistance of the strain.

### 2.3. Evaluation of Bacterial Metabolic Activity by Metabolizing Resazurin

The results of the Resazurin standardization using the McFarland scale showed that the higher the concentration of the bacteria, the higher was the metabolism of the reagent, which may suggest that bacterial viability and bacterial growth are proportional, as shown in Figure 5.

In all the results of the Resazurin metabolization, the fluorescence signal was subtracted only from the reagent in both strains. In the MSSA strains, after the application of aPDT, the absorbance of the control group was only irradiated and the samples treated only with the PS showed a similar pattern of metabolization, indicating that there was no reduction in the viability of these groups. The groups that received aPDT presented a significant reduction in their viability (*p* < 0.05) compared to the controls.

The non-irradiated and control groups obtained absorbance values around 900 to 1000 (a.u.), which corresponds approximately to 10 in the McFarland scale. The groups submitted to aPDT obtained absorbance values around 0 (a.u.), confirming that there was no bacterial metabolism, except in the parameters of 25 J/cm^2^ and 25 mg/L, where it was possible to observe a slight metabolic activity, as shown in Figure 6.

As observed in the CFU/mL analysis, the pattern of response to aPDT of the MSSA and MRSA strains was similar when it was analyzed through Resazurin metabolization (Figure 6 and Figure 7). In MRSA strains, the control and non-irradiated groups showed absorbance around 1000 (a.u.), corresponding to 10 in the McFarland scale. The aPDT groups obtained absorbance around 0 (a.u.), confirming a marked decrease in bacterial metabolism, as shown in Figure 7.

### 2.4. Evaluation of the Production of ROS.

The amount of ROS was quantified to relate ROS production with the effectiveness of aPDT in the studied strains. The irradiations of 25, 50, and 100 J/cm^2^ from the previous experiments were maintained.

In the MSSA strains, an increase in ROS production of approximately 20 a.u. was observed for the groups irradiated at 25 and 50 J/cm^2^ in all PDZ concentrations in relation to the light control groups (LED) (*p* < 0.05), except for the groups with parameters of 25 and 50 mg/L and 25 J/cm^2^, in which there were no significant differences. Regarding only the fluences of 25 and 50 J/cm^2^, the PDZ concentrations of 100 and 75 mg/L obtained the highest ROS production (Figure 8).

The groups irradiated at 100 J/cm^2^ exhibited a growth of approximately 130 a.u. in relation to The LED group at all concentrations (*p* < 0.05), reaching the highest ROS production at 75 mg/L.

In the MRSA strains, regarding only fluences of 50 and 25 J/cm^2^, there was a similar production of ROS at 100, 75, and 50 mg/L (Figure 9), with differences of approximately 20 a.u. between all LED groups (*p* < 0.05). A lower production was observed at 25 mg/L, with no significant differences being observed in the group with the parameters of 25 mg/L and 25 J/cm^2^.

Regarding the MSSA strain, there was a greater increase in the production of ROS in the groups irradiated at 100 J/cm^2^, with an average difference of approximately 160 a.u. compared to the LED group at all concentrations (*p* < 0.05), highlighting the highest ROS production at 50 mg/L.

Regarding the 100 J/cm^2^ fluence, the ROS production has increased for both strains, reaching higher amounts of ROS compared to that obtained under the fluences of 25 and 50 J/cm^2^. Therefore, in this case, ROS production was fluence dependent, that is, the higher the fluence applied, the higher the ROS production.

## 3. Discussion

One of the points evaluated in this study was the internalization of PDZ in the MSSA and MRSA strains, in which high interaction and internalization of the PS were observed at all tested concentrations. In 2018, Pereira et al. performed an internalization analysis for strains of *S. aureus* using methylene blue at 100, 300, and 500 mg/L, with an incubation time of 15 min. The results showed that methylene blue was internalized in the *S. aureus* strains. When evaluating bacterial inactivation after the application of aPDT with a fluence of 25 J/cm^2^, the reduction was of approximately 4 logs in relation to the control, not resulting in complete inactivation, as observed in this study using the PDZ as PS [11]. Such results demonstrate that, despite internalization, the type of compound used as PS may directly affect the success of aPDT, not depending only on the internalization of this PS into the bacterial cells to cause inactivation of the strains. In addition, the similarity in the parameters (100 mg/L and 25 J/cm^2^) used in both studies reinforces the fact that the factor that triggered the difference in the results was the PS used, requiring future studies to understand how the PS structures may affect the interaction with the bacteria to generate ROS.

PS absorption by the bacteria is one of the main factors responsible for the effectiveness of the photodynamic inactivation process. The present study showed that 15 min of PS incubation were sufficient to guarantee internalization in the strains at all concentrations used and to result in total bacterial inactivation in most of the parameters employed. In 2017, Winkler et al. applied aPDT to MRSA strains, varying the incubation period in 5, 15, 30, 60, and 90 min, the concentrations of Chlorine e6 (Ce6) varied from 2 to 1024 μM, and the fluences were 1.86, 9.3, and 18.6 J/cm^2^. Regarding the MRSA 44 strain, Winkler et al. (2017) reported that among all incubation times there were no significant differences in the internalization of PS in bacterial cells, suggesting that shorter incubation times were sufficient for internalization to occur. Still, to prove this hypothesis, they tested three genetically independent MRSA isolates at the aforementioned concentrations, with 30 min of incubation, and observed that there were differences in the intracellular accumulation of PS at higher concentrations (≥128 μM), suggesting that differences in the incubation period were related to the higher concentrations used by them, being an important factor that affects the accumulation of Ce6 by *S. aureus*, and consequently, the generation of ROSs [20].

Assessing the variation in light fluences, as in the present study, Winkler and collaborators observed that the decrease in the irradiation time affected the efficiency of aPDT. Nevertheless, the highest light flow used by them is lower than the light flow most used in the present study, reporting that lower light streams with high concentrations of Ce6 result in a bacterial reduction. Moreover, Winkler et al. tested aPDT with equal parameters in 12 clinical isolates of *S. aureus* with different genetic and resistance patterns, obtaining the same result with a reduction of 5 logs for all groups, reporting that the effectiveness of aPDT, regardless of the resistance pattern of the strains [20], corroborated the results of the present study, in which there were similarities in the results of both the MSSA and MRSA strains, regardless of their methicillin resistance.

This study demonstrated, in vitro, that the complete inactivation of the MSSA and MRSA strains was possible using PDZ as PS. This factor may be related to the molecular properties of this compound. In 2015, Quishida et al. obtained a 3–4 log reduction in *Streptococcus mutans* biofilms using PDZ at concentrations of 175 and 200 mg/L and irradiation of 37.5 J/cm^2^ [21]. As *S. aureus* and *S. mutans* are two Gram-positive bacteria, these results demonstrate that PDZ has a high potential to cause the inactivation of Gram-positive bacteria, which might be explained by the cell wall structure of these microorganisms. Gram-positive bacteria have dense layers of peptidoglycan interspersed with the formation of teichoic acids in their cell wall, which is a porous structure, with low electrical charge and does not represent an obstacle for the passage of small particles [22,23,24].

PDZ is a Ce6 modified by the addition of N-methyl-D-glucosamine, a solubilizing and stabilizing agent which facilitates its penetration into cells. In addition, it has molecular groups at its ends that may charge the PS, which can be a relevant factor in its interaction with the Gram-positive bacteria [23,25]. The PS aggregation decreases the potential to generate a singlet oxygen since the aggregates contribute to reducing the action time of the singlet and triplet states, affecting the cytotoxic effect responsible for the inactivation of the microorganisms [26]. In 2012, Correia et al. found that the application of PDZ has advantages when compared with a hematoporphyrin-derived PS (Photogen^®^) as it presents a higher absorption in the red light, does not present aggregations at a concentration of 120 mg/L, and eliminate faster the compound from the body, reducing the photosensitivity effect of the skin [27]. Based on the characteristics presented, and on the results of complete inactivation obtained in the present study, it is possible to state that aPDT with PDZ is a potential treatment for combating infections by MRSA and MSSA strains.

Although this study deals with planktonic bacteria, the study of Quishida et al. demonstrated the potential of PDZ against the biofilm of a Gram-positive bacterium, suggesting the potential for future inactivation of the *S. aureus* biofilm, associated with the results of the present study. However, according to Perussi (2007), some factors may interfere with the response to aPDT between species with the same Gram classification, such as divergences between the size of the microbial cell, membrane permeability barriers, differences in antioxidant enzymes in DNA, or DNA repair mechanisms, requiring further studies to prove this hypothesis [21,25].

Garcia and colleagues (2018) reported the action of aPDT on biofilms of *Streptococcus mutans* using three PSs: PDZ (600 μg and Fotoenticine^®^ (FTC) (600 μg/mL), chlorine derivatives, and the methylene blue (MB) (1000 μg/mL). These PSs were submitted to laser irradiation with a fluence of 39.5 J/cm^2^ (660 nm), presenting CFU/mL reductions of 4 log10 with MB, 6 log10 with PDZ, and total inactivation of the biofilm with FTC. Although FTC has achieved complete inactivation, both chlorine-derived PSs have exhibited higher antimicrobial activity against *S. mutans* than AM, indicating that these PSs may be effective in the inactivation of Gram-positive bacteria [18]. In this study, the application of aPDT in the Gram-positive bacterium *S. aureus* on a planktonic form has achieved total inactivation with PDZ concentrations lower than those found in the study of Garcia et al., requiring future studies to demonstrate the potential use of these parameters in the form of biofilm, to determine if a more virulent strain is affected by aPDT with PDZ. The present study revealed that MRSA strains, considered more virulent due to their methicillin resistance, were more susceptible to aPDT, while MSSA strains, which are sensitive to this antibiotic, presented resistance to the therapy. This difference was observed in the parameters of 50 mg/L and 25 J/cm^2^ and 25 mg/L and 50 J/cm^2^ and was reflected in the number of CFU/mL, which presented some growth, while under the same parameters there was total bacterial inactivation in the MRSA strains. This result is remarkable, since this reduced number of surviving bacteria would be easily controlled by the immune system of the host, thus preventing the infection from advancing. In this sense, a study by Freitas et al. (2019) applied aPDT to MRSA and MSSA strains using curcumin (100 mg/L) with irradiation of 8 and 20 J/cm^2^ at a wavelength of 450 nm as PS. Regarding CFU/mL there was a reduction of approximately 4 log10 of CFU/mL for all treated groups, with no significant differences between the clinical strains, corroborating the results of the present study, in which aPDT was effective, regardless of resistance or sensitivity to antibiotic [28]. Moreover, ROS production increased significantly after the application of aPDT in the tested strains. They used two light fluences (8 and 20 J/cm^2^), with the higher one resulting in the generation of more ROSs, confirming the results of our study, in which ROS production was dependent on the intensity of light fluence [28].

Although a small number of strains have been used in this work, a study by Sabino et al. (2020) reported that sensitive and resistant strains belonging to the same bacterial species present a similar sensitivity to aPDT [29]. This observation can be explained by the way antibiotic resistance mechanisms work in bacteria. In the case of *S. aureus*, three mechanisms have been described that explain resistance to antibiotics and β-lactam: (i) production of inactivating enzymes (and β-lactams); (ii) modification of penicillin-binding proteins (PBPs); and (iii) intrinsic resistance to methicillin, the latter being the most relevant. The first is activated by the presence of an antibiotic, such as penicillin, which induces the synthesis of enzymes that lead to its inactivation. On the other hand, the other mechanisms express PBP enzymes that are not significantly affected by the antibiotic, continuing with the synthesis of the bacterial wall without loss of viability. Bearing this in mind, in the absence of the β-lactam antibiotic, the MRSA strain will not induce the production of β-lactams and its wall synthesis will be similar to that of the MSSA strain, which would explain why aPDT is effective in both strains. Finally, small differences in the viability of *S. aureus* strains after aPDT can be explained by the innate capacity (production of antioxidants) of each strain to eliminate ROS [30,31,32]. Therefore, in future work, it will be important to evaluate a higher number of microorganisms.

The resazurin test used in this study showed a decrease in the bacterial metabolic activity at all concentrations in the groups in which the therapy was applied, with the light fluences used reaching 0 (a.u.), whereas in the control and in the only irradiated groups, we observed relative fluorescence at concentrations higher than the initial concentration of the experiment, demonstrating the efficacy of aPDT in reducing the bacterial metabolic activity. Germ et al. (2019) and Jia et al. (2020), analyzed bacterial resistance to the antibiotic colistin in Gram-negative strains using the resazurin test, determining the level of cell viability, correlating it with the level of resistance to the antibiotic used [33,34]. Mishra et al. and Ravi et al., in 2019, also used resazurin to assess bacterial metabolism, concluding that in addition to being a quick and easy method, it can contribute to the rapid selection of antibiotics and, consequently, to reduce the progression of bacterial infection [35,36]. The present study demonstrated the effectiveness of aPDT by showing a direct relationship between resazurin and CFU/mL tests, in which the higher the bacterial concentration, the greater the reagent consumption and, consequently, the greater the bacterial metabolic activity. We suggest that the resazurin test might be considered an option to measure the bacterial concentration in a short period of time as the CFU/mL test takes 24 h to generate a complete result, while the resazurin test was able to provide information within 4 h. However, it is important to note that it is a less sensitive test than the CFU/mL and, although it can relate the small growth and metabolic activity for the MRSA strain in the parameters of 25 J/cm^2^ and 25 mg/L, it was not possible to quantify colonies and observe differences, for example, between the aPDT groups of 25 and 50 J/cm^2^ for the MSSA strains, in which there was slight bacterial growth in the CFU/mL test.

The excessive production of intracellular ROS triggered by aPDT is a factor that can result in cell death, destroying the cytoplasmic membrane and causing fatal damage by the intracellular oxidation of nucleic acids and proteins [37]. In this study, the DCFDA marker was used to detect the level of ROS after therapy. According to the manufacturer, after diffusion in the cell, the compound is deacetylated by cellular esterases in a non-fluorescent compound (DCFH) and later oxidized by the ROS in 2′, 7′—dichlorofluorescein (DCF), which is the fluorescent compound that directly reflects the total ROS level within the bacteria. In 2019, Mao et al. evaluated the production of ROS with DCFDA and observed that the *S. aureus* strains that were submitted to aPDT yielded twice more ROS than the control group. Moreover, associated with photodynamic and photothermal therapy, they obtained a production of ROS four times higher. Regarding clinical application, photothermal therapy at an elevated temperature could be harmful to patients. However, Mao et al. (2019), reported that high temperature can damage the bacterium membrane, resulting in higher internalization of the PS and consequently in more ROS, therefore being interesting factors for future studies, in which treatments with high temperatures may also prove beneficial. Mao et al. also evaluated the production of singlet oxygen and hydroxyl radicals separately, observing higher production after irradiation, concluding that the small amount of ROS generated in the control group belongs to the species usually produced by the normal bacterial aerobic respiration, as well as the ROS produced in the LED group of the present study [38].

The comparison of this investigation with Mao et al. (2019) showed that ROS production also increased significantly after the application of aPDT and that, in this protocol, the use of high-intensity light streams was not necessary to generate the amount of ROS required to inactivate completely all the bacteria. When comparing ROS production with CFU/mL, complete bacterial inactivation occurred at 50 mg/L and 25 J/cm^2^ for the MRSA strain and 50 mg/L, and 50 J/cm^2^ for the MSSA strain, in which low arbitrary units (a.u.) were generated when compared to a fluence of 100 J/cm^2^, resulting in high amounts of a.u. at all concentrations.

As low flows of light can have the same inactivating effect as the high fluences, it is assumed that the low flows used were sufficient to excite the PDZ, generating a sufficient amount of ROSs. Therefore, in this case, this study suggests the use of low-intensity light streams to reduce costs and achieve a shorter exposure time of the patient to the treatment, therefore avoiding adverse effects that can be caused by prolonged irradiation in the tissues adjacent to the lesion. However, it is important to emphasize that investigations on each strain are necessary to find the appropriate threshold in ROS production, since besides depending on the fluency of light used, the ideal PS needs to be found, as well as the appropriate concentrations for each bacterial species to result in their complete inactivation.

## 4. Materials and Methods

### 4.1. Obtaining and Maintaining Bacterial Strains

The clinical MSSA and MRSA strains used in the present study were obtained in a partnership with the Oswaldo Cruz Laboratory of São José dos Campos and are part of the collection of the Laboratory of Photobiology Applied to Health. The strains were kept frozen at −20 °C in suspension with Brain heart Infusion broth (BHI) with 5% glycerol [39].

### 4.2. Photosensitizer Preparation

The photosensitizer Photodithazine^®^—Fotoditazin^®^, Veta-Grand, Russia, provided by the Optics Group of the São Carlos Institute of Physics—USP, is produced by the Russian company Veta-Grand^®^. It was kept in a 5 mg/mL stock solution and stored at 4 °C, being diluted in Phosphate Buffered Saline (PBS) at 25, 50, 75, and 100 mg/L for experimental use.

### 4.3. Bacteria Preparation

The strains used in this study were obtained from the Oswaldo Cruz Laboratory, São José dos Campos, Brazil. The solutions of *S. aureus* were cultivated in a BHI broth (Brain Heart Infusion), followed by incubation for 24 h at 37 °C. After growth, the tubes were centrifuged for 15 min at 3000 rpm, then the discarded supernatant was added to the sterile PBS until obtaining a bacterial solution equivalent to the 0.5 McFarland scale (1.5 × 10^8^ cells/mL) [39].

### 4.4. Cytotoxicity analysis of Photodithazine^®^

To assess the cytotoxicity of PDZ in the dark, tests were carried out at 25, 50, 75, and 100 mg/L (PDZ groups). The experimental groups were incubated with 100 µL of the PS dilution for 15 min. After incubation, 10 µL of these cultures were seeded into the BHI agar plates and incubated at 37 °C for 24 h. Finally, the colonies were counted and Colony Forming Units (CFU/mL) were calculated [28].

### 4.5. Irradiation

The aPDT was performed in triplicate with the PDZ groups being subjected to irradiation with fluence at 25, 50, and 100 J/cm^2^ and with the control groups kept in the absence of light and only irradiated without the PDZ. The irradiation was performed using Biotable (Biotable/ Biopdi660), being composed of 54 LEDs with a wavelength of 660 nm, power of 70 mW, the irradiance of 25 mW/cm^2^, and irradiation times of 16 min and 40 s (25 J/cm^2^), 33 min and 20 s (50 J/cm^2^), and 1 h and 6 min and 40 s (100 J/cm^2^), respectively.

### 4.6. Internalization of Photodithazine^®^ in the MSSA and MRSA Strains with Confocal Microscopy

To assess the internalization of PDZ in bacteria, the strains were diluted in a concentration equivalent to 10 in the McFarland scale (3 × 10^9^ cells/ mL) and centrifuged at 3500 rpm for 10 min. Then, PDZ was added at 100, 75, 50, and 25 mg/L and incubated for 15 min. The suspension was again centrifuged, the PDZ was removed, and the bacteria fixed with 4% paraformaldehyde and adhered to coverslips previously treated with Poli-L-lysine. After 24 h, the slides were mounted and analyzed under an LSM 700 Zeiss Confocal Microscope with excitation for DAPI (4′,6-Diamidino-2-Phenylindole, Dihydrochloride) at 405 nm and PDZ at 555 nm [11].

### 4.7. Colony Forming Units Count (CFU/mL)

Initially, the MSSA and MRSA strains were diluted at 0.5 in the McFarland scale, then incubated with PDZ at 25, 50, 75, and 100 mg/L for 15 min. The irradiation of cultures was performed in 24-well plates. After applying the aPDT, the experimental groups were diluted, sown on BHI agar, and incubated at 37 °C for 24 h. After incubation, the number of Colony Forming Units (CFU/mL) was counted, with the results being expressed on a logarithmic scale (Log_10_) [11].

### 4.8. Analysis of ROS

The analysis of ROS includes an assessment of singlet oxygen and free radicals. Was performed with the reagent 2′diacetate marker, 7′-dichlorodihydrofluorescein (DFCDA), a reduced form of fluorescein used as an indicator for ROS in cells. Before irradiation at fluences of 25, 50, and 100 J/cm^2^, the reagent was added and incubated for 30 min. The reading was performed on the fluorescence spectrophotometer with 530 nm emission and at 480 nm excitation, obtaining the results in fluorescence [28].

### 4.9. Assessment of Bacterial Metabolism Through Resazurin Metabolization

The assessment of metabolic activity with resazurin is a fluorometry test capable of determining the bacterial metabolic activity. In this test, viable and metabolically active bacteria metabolize the blue reagent, generating a pink color, while non-viable bacterial cells do not produce such a reaction. To experiment, a sterile stock solution of Resazurin at 6.75 mg/mL was prepared and stored at −4 °C in the dark.

Standardization was carried out to assess its viability and compare it to the growth of MSSA and MRSA strains after aPDT, according to the 0.5 to 10 scores in the McFarland scale. The bacterial suspensions were incubated with 2 µL of resazurin at 6.75 mg/mL, and the incubation period lasted for 4 h at 37 °C. After that, the absorbance was read with a fluorescence spectrophotometer with excitation at 528 nm and emission at 645 nm.

After the application of aPDT, the viability of the MRSA and MSSA strains with resazurin was evaluated using the same parameters used for the standardization. However, post-aPDT bacterial solutions were used to compare bacterial viability before and after therapy application [39].

### 4.10. Statistical Analysis

The results obtained were submitted to statistical analysis using the Bioestat 5.0 software, with ANOVA and Tukey tests, considering a value of *p* < 0.05 as statistically significant [5].

## 5. Conclusions

The results showed that PS Photodithazine^®^ at 25, 50, 75, and 100 mg/L is not cytotoxic for MSSA and MRSA strains, and when it is submitted to aPDT at fluences of 100, 50, and 25 J/cm^2^ can reduce bacterial viability, with complete bacterial inactivation at 100 and 75 mg/L under all light wavelengths used in the two treated strains. In addition, by associating a lower concentration of PS and lower fluences until complete bacterial inactivation, 50 mg/L and 25 J/cm^2^ were obtained for the MRSA strain, and 50 mg/L and 50 J/cm^2^ for the MSSA strain, with these concentrations being recommended for possible clinical application.

The therapy was effective regardless of the resistance or sensitivity of the strains to the antibiotic methicillin, confirming the potential of Antimicrobial Photodynamic Therapy as an alternative in the treatment of superficial bacterial infections by *Staphylococcus aureus,* without affecting microbial resistance.

## Figures and Tables

**Figure 1 antibiotics-10-00869-f001:**
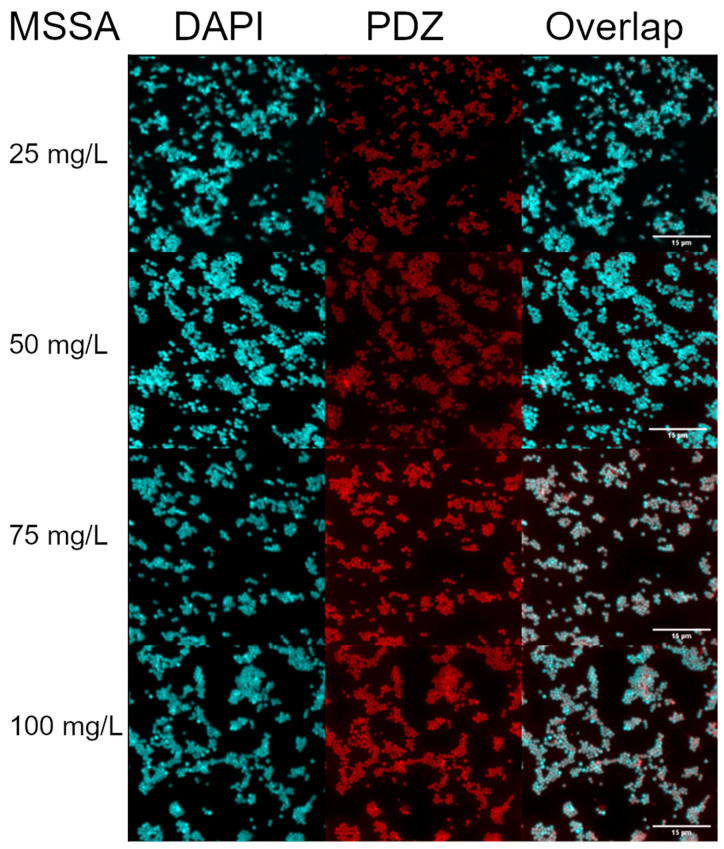
Internalization of PDZ in MSSA strains at 100, 75, 50, and 25 mg/L after 15 min of incubation, observed under Confocal Microscopy. The marking of the bacterial DNA with DAPI is observed in blue, and the PDZ interaction with bacterial strains, in red, as well as the overlapping of signals.

**Figure 2 antibiotics-10-00869-f002:**
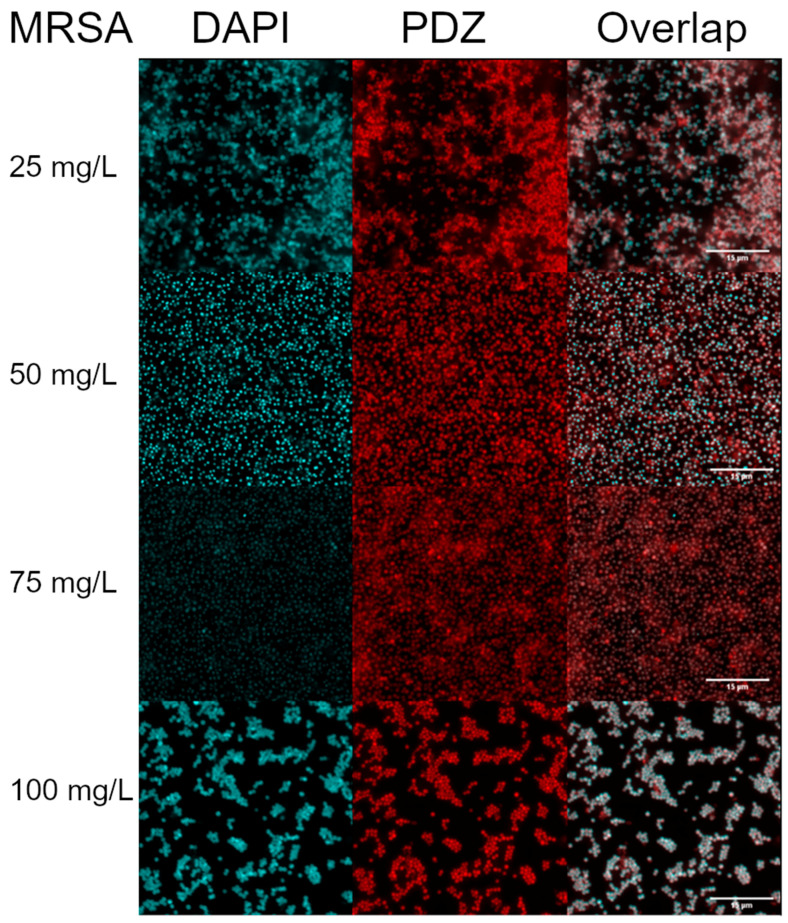
Internalization of PDZ in MRSA strains at 100, 75, 50, and 25 mg/L after 15 min of incubation, observed under Confocal Microscopy. The marking of the bacterial DNA with DAPI is observed in blue, and the PDZ interaction with bacterial strains in red, as well as the overlapping of signals.

**Figure 3 antibiotics-10-00869-f003:**
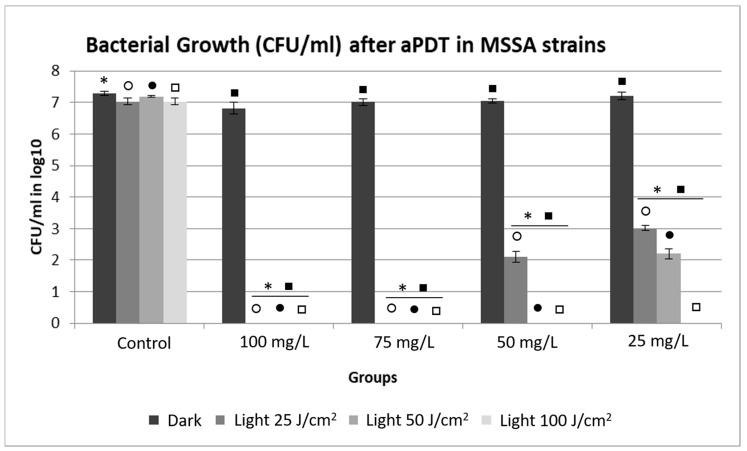
Average values of CFU/mL in log10 for MSSA strains, showing a Dark and Light control group, non-irradiated Photodithazine^®^ at 25, 50, 75, and 100 mg/L and irradiated at 25, 50, and 100 J/cm^2^. * Significant differences were observed between the Dark Control group to 100, 75, 50, and 25 mg/L with Dark, light 25, 50, and 100 J/cm^2^ groups. ○ Significant differences were observed between the light 25 J/cm^2^ Control group to 100, 75, 50, and 25 mg/L with light 25 J/cm^2^ groups. ● Significant differences were observed between the light 50 J/cm^2^ Control group to 100, 75, 50, and 25 mg/L with light 50 J/cm^2^ groups. □ Significant differences were observed between the light 100 J/cm^2^ Control group to 100, 75, 50, and 25 mg/L with light 100 J/cm^2^ groups. ■ Significant differences were observed between the Dark 100, 75, 50, 25 µl/mL groups to 100, 75, 50, 25 mg/L with light 100, 75, 50, and 25 J/cm^2^ groups.

**Figure 4 antibiotics-10-00869-f004:**
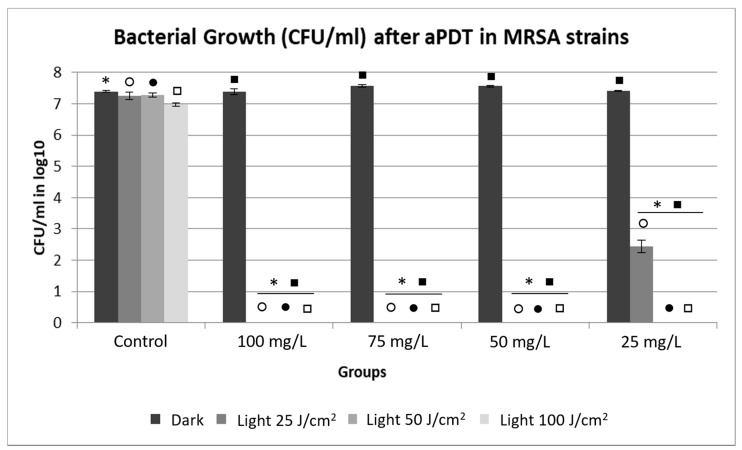
Average values of CFU/mL in log10 for MRSA strains showing Dark and Light control groups; Photodithazine^®^ at 25, 50, 75, and 100 mg/L; and non-irradiated and irradiated groups with fluences of 25, 50, and 100 J/cm^2^. * Significant differences were observed between the Dark Control group to 100, 75, 50, and 25 mg/L with Dark, light 25, 50, and 100 J/cm^2^ groups. ○ Significant differences were observed between the light 25 J/cm^2^ Control group to 100, 75, 50, and 25 mg/L with light 25 J/cm^2^ groups. ● Significant differences were observed between the light 50 J/cm^2^ Control group to 100, 75, 50, and 25 mg/L with light 50 J/cm^2^ groups. □ Significant differences were observed between the light 100 J/cm^2^ Control group to 100, 75, 50, and 25 mg/L with light 100 J/cm^2^ groups. ■ Significant differences were observed between the Dark 100, 75, 50, 25 mg/L groups to 100, 75, 50, and 25 mg/L with light 100, 75, 50, and 25 J/cm^2^ groups.

**Figure 5 antibiotics-10-00869-f005:**
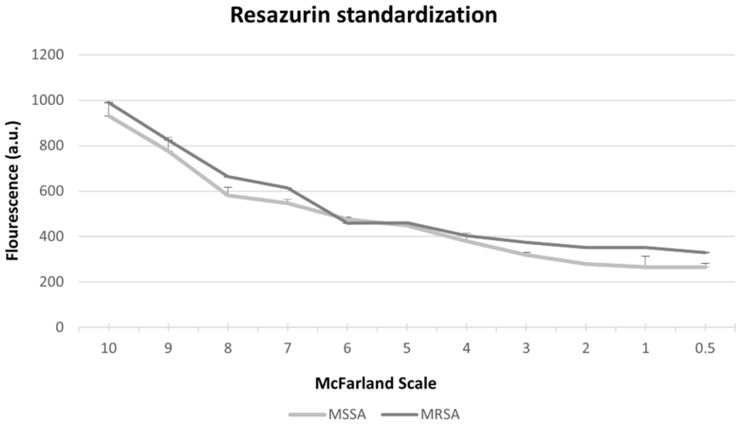
Absorbance of Resazurin on the McFarland scale (0.5 to 10) in the MSSA and MRSA strains.

**Figure 6 antibiotics-10-00869-f006:**
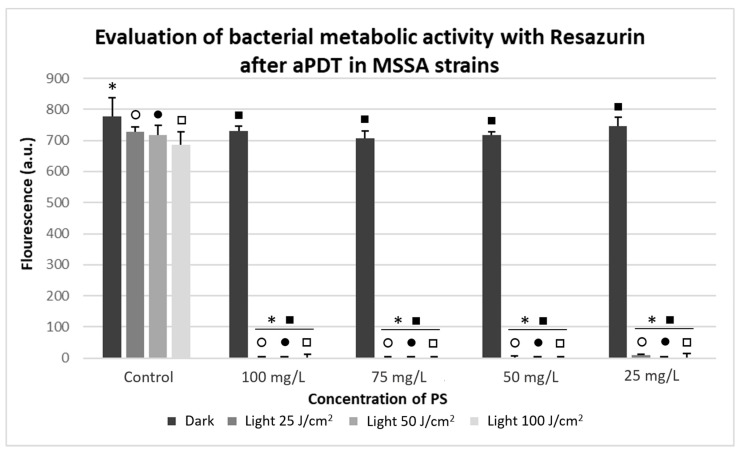
Evaluation of bacterial metabolic activity with Resazurin in arbitrary fluorescence units for MSSA strains, showing Dark and Light control groups; Photodithazine^®^ at 25, 50, 75, and 100 mg/L; and non-irradiated and irradiated groups with fluences of 25, 50, and 100 J/cm^2^. * Significant differences were observed between the Dark Control group to 100, 75, 50, and 25 mg/L with Dark, light 25, 50, and 100 J/cm^2^ groups. ○ Significant differences were observed between the light 25 J/cm^2^ Control group to 100, 75, 50, and 25 mg/L with light 25 J/cm^2^ groups. ● Significant differences were observed between the light 50 J/cm^2^ Control group to 100, 75, 50, and 25 mg/L with light 50 J/cm^2^ groups. □ Significant differences were observed between the light 100 J/cm^2^ Control group to 100, 75, 50, and 25 mg/L with light 100 J/cm^2^ groups. ■ Significant differences were observed between the Dark 100, 75, 50, 25 mg/L groups to 100, 75, 50, and 25 mg/L with light 100, 75, 50, and 25 J/cm^2^ groups.

**Figure 7 antibiotics-10-00869-f007:**
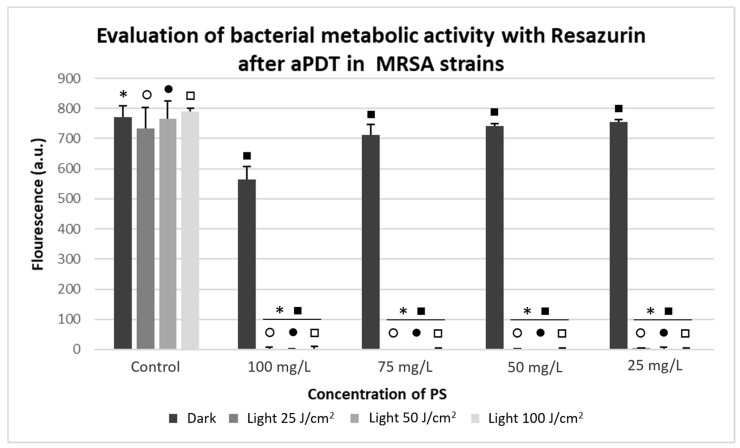
Evaluation of bacterial metabolic activity with Resazurin in arbitrary fluorescence units for MRSA strains, showing Dark and Light control groups, Photodithazine^®^ at 25, 50, 75, and 100 mg/L; and non-irradiated and irradiated groups with fluences of 25, 50, and 100 J/cm^2^. * Significant differences were observed between the Dark Control group to 100, 75, 50, and 25 mg/L with Dark, light 25, 50, and 100 J/cm^2^ groups. ○ Significant differences were observed between the light 25 J/cm^2^ Control group to 100, 75, 50, and 25 mg/L with light 25 J/cm^2^ groups. ● Significant differences were observed between the light 50 J/cm^2^ Control group to 100, 75, 50, and 25 mg/L with light 50 J/cm^2^ groups. □ Significant differences were observed between the light 100 J/cm^2^ Control group to 100, 75, 50, and 25 mg/L with light 100 J/cm^2^ groups. ■ Significant differences were observed between the Dark 100, 75, 50, 25 mg/L groups to 100, 75, 50, and 25 mg/L with light 100, 75, 50, and 25 J/cm^2^ groups.

**Figure 8 antibiotics-10-00869-f008:**
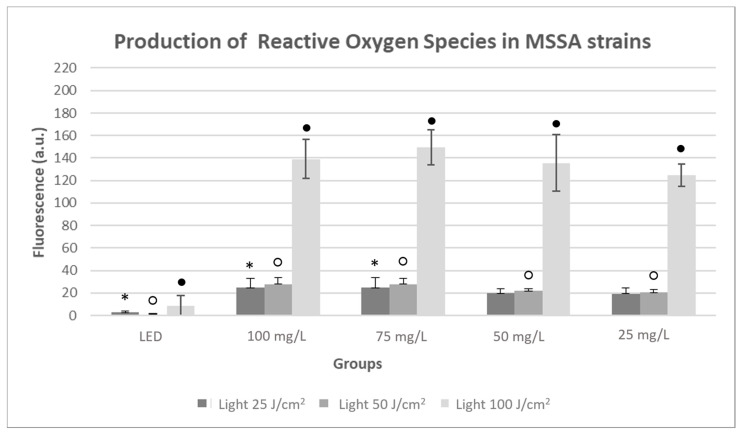
Production of ROS in MSSA strains, presenting a Light control group without PS (LED), Photodithazine^®^ at 25, 50, 75, and 100 mg/L, all irradiated (Light) at the fluences of 25, 50, and 100 J/cm^2^. * Significant differences were observed between the LED group to 100 and 75 mg/L with light 25 J/cm^2^ groups;_._ ○ Significant differences were observed between the LED group to 100, 75, 50, and 25 mg/L with light 50 J/cm^2^ groups; ● Significant differences were observed between the LED group to 100, 75, 50, and 25 mg/L with light 100 J/cm^2^ groups.

**Figure 9 antibiotics-10-00869-f009:**
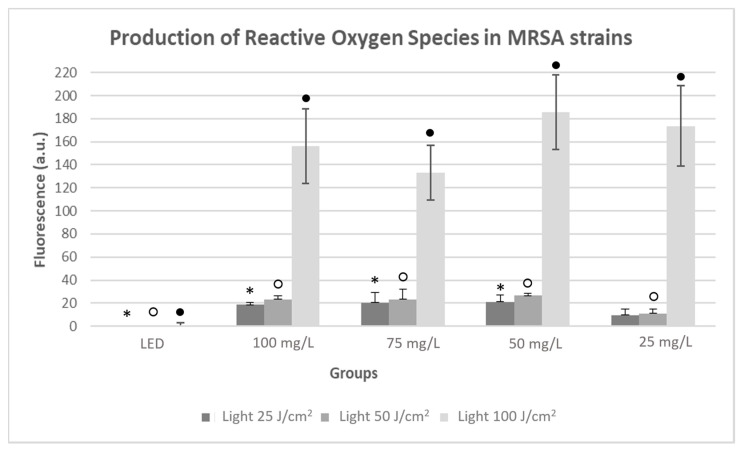
Production of ROS by MRSA strains, with a Light control group without PS (LED); Photodithazine^®^ at 25, 50, 75, and 100 mg/L; all irradiated (Light) at the fluences of 25, 50, and 100 J/cm^2^. * Significant differences were observed between the LED group to 100, 75, and 50 mg/L with light 25 J/cm^2^ groups. ○ Significant differences were observed between the LED group to 100, 75, 50, and 25 mg/L with light 50 J/cm^2^ groups_._ ● Significant differences were observed between the LED group to 100, 75, 50, and 25 mg/L with light 100 J/cm^2^ groups.

## Data Availability

Not applicable.

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
