# Peer review of "Efficiency of Antimicrobial Photodynamic Therapy with Photodithazine^®^ on MSSA and MRSA Strains"

_antibiotics, 2021, doi:10.3390/antibiotics10070869_

Round 1

Reviewer 1 Report

The manuscript presents an original research study. The improvement of the presentation needed as followed:

  1. Figs. 3 and 4 Please use the log Y-scale. As presented it looks like the results are not appropriate.
  2. the concentrations in microgram per mL are too high. Maybe per L?
  3. Fig. 8 Singlet oxygen by 1270 nm or another property? Not clear.
  4. Give the structure of the used photosensitizer, the absorption spectrum and the light sourse parameters.
  5. Overal re-writing of the whole manuscript in order to be clear for the readers.

Author Response

Response to  reviewers

The authors appreciate the valuable points from the reviewers. We believe that the changes suggested and made have made the manuscript clearer. We hope that the changes described below will improve the article.

Reviewer 01:

The manuscript presents an original research study. The improvement of the presentation needed as followed:

  • 1) Figs. 3 and 4 Please use the log Y-scale. As presented it looks like the results are not appropriate.

Response: The both figures were alterated.

  • 2) the concentrations in microgram per mL are too high. May be per L?

Response: The units were changed to mg/L;

  • 3) Fig. 8 Singlet oxygen by 1270 nm or another property? Not clear.

Response: The purpose of the test using the probe 2 'diacetate marker, 7'-dichlorodihydrofluorescein (DFCDA) is to quantify, by means of fluorescence, reactive oxygen species. This is done after the interaction of the groups with the probe, and excitation at 480 nm and emission reading at 528 nm, following previous experiments. No specific quantification of singlet oxygen was performed. The information was added in the MM.

  • 4) Give the structure of the used photosensitizer, the absorption spectrum and the light source parameters.
  • Response:
  •  
  • Source: Bruno Andrade Ono. Evaluation of the photodynamic response in murine melanoma cells using Photodithazine. Dissertation presented to the Graduate Program in Physics at the São Carlos Institute of Physics. 2016.
  •  
  • 5) Overal re-writing of the whole manuscript in order to be clear for the readers.

Response: We believe that the changes made after the reviewers' suggestions throughout the manuscript have made it clearer.

Reviewer 2 Report

General:

do not capitalize Methicillin-sensitive and Methicillin-resistant, only when they are at the beginning of the sentence.

methicillin and other antibiotic names should NOT be capitalized

in vitro, in vivo should be in italics, Gram-positive and Gram-negative (with hyphen)

L11: associated with high mortality rates

L17: 25, 50 and 100

please write Reactive Oxygen Species on first mention, and later on: ROS

L31: Please consider including the following reference:

https://pubmed.ncbi.nlm.nih.gov/18318877/

https://pubmed.ncbi.nlm.nih.gov/33799337/

L39: please use a different word for retraction

Define MRSA on the first mention in the text

L45: pre-clinical

L48-L51: please consider discussing the possible utility of photodynamic therapy in various medical/therapeutic areas, see reference:

https://pubmed.ncbi.nlm.nih.gov/32392793/

L57: is everything correct within the sentence? Spirulina should be in italics

L67: coconut aggreement? is that the most appropriate description?

All figures: „Source: Author” is not necessary

L84: ≤0.05 is inappropriate. either below (meaning statistical significance) or equal and above (no stat. significance), throughout the manuscript!

Figure 3 and 4, 6. Please explain the singificance of the boxes, stars and other legends…

Discussion:

the current state of the discussion is quite extensive, but I would like to see some discussion of the role of photodynamic therapy in the elimination of biofilm-embedded MSSA/MRSA and biofilm-related infections in general.

Statistical analysis: same issue with p value, see above!

Author Response

Reviewer 02:

General:

Do not capitalize Methicillin-sensitive and Methicillin-resistant, only when they are at the beginning of the sentence. methicillin and other antibiotic names should NOT be capitalized

Response: It was changed according to the reviewer's suggestion.

in vitro, in vivo should be in italics, Gram-positive and Gram-negative (with hyphen)

Response: It was changed according to the reviewer's suggestion.

L11: associated with high mortality rates

Response: It was changed according to the reviewer's suggestion.

L17: 25, 50 and 100

Response: It was changed according to the reviewer's suggestion.

please write Reactive Oxygen Species on first mention, and lateron: ROS

Response: It was changed according to the reviewer's suggestion.

L31: Please consider including the following reference:

https://pubmed.ncbi.nlm.nih.gov/18318877/

https://pubmed.ncbi.nlm.nih.gov/33799337/

Response: We appreciate the suggestion. References have been added to the text..

L39: please use a different word for retraction

Response: “Retraction” has been replaced by “reduction”. Thank you for the observation.

Define MRSA on the first mention in the text

Response: It was changed according to the reviewer's suggestion.

L45: pre-clinical

Response: It was changed according to the reviewer's suggestion.

L48-L51: please consider discussing the possible utility of photodynamic therapy in various medical/therapeutic areas, see reference:

https://pubmed.ncbi.nlm.nih.gov/32392793/

Response: The other applications of Photodynamic Therapy were mentioned in the introduction. The reference has been added, thanks.

L57: is everything correct within the sentence? Spirulina should bein italics

Response: It was changed according to the reviewer's suggestion.

L67: coconut aggreement? is that the most appropriatedescription?

Response: It was changed. The “coconut agreement” was replaced by “grape-like clusters”.

All figures: „Source: Author” is not necessary

Response: It was changed according to the reviewer's suggestion.

L84: ≤0.05 is inappropriate. either below (meaning statisticalsignificance) or equal and above (no stat. significance), throughoutthe manuscript!

Response: It was changed according to the reviewer's suggestion. All “p: ≤0.05” have been changed to “p <0.05

Figure 3 and 4, 6. Please explain the singificance of the boxes,stars and other legends…

Response: It was changed according to the reviewer's suggestion. The significance was added in the figure caption.

Discussion:

the current state of the discussion is quite extensive, but I wouldlike to see some discussion of the role of photodynamic therapy inthe elimination of biofilm-embedded MSSA/MRSA and biofilm-related infections in general.

Response: It was changed according to the reviewer's suggestion. The discussion was shortened. Our group is currently working on projects specifically with biofilms, based on these initial studies in planktonic cultures that we hope will be published soon.

Statistical analysis: same issue with p value, see above!

Response: It was changed according to the reviewer's suggestion.

Round 2

Reviewer 2 Report

The authors have adequately addressed all concerns during the revision.